# Harnessing non-equilibrium forces to optimize work extraction

Kristian Stølevik Olsen [1] ✉, Rémi Goerlich[1,2], Yael Roichman[2,3] & Hartmut Löwen[1]

While optimal control theory offers effective strategies for minimizing energetic costs in noisy microscopic systems over finite durations, a significant opportunity lies in exploiting the temporal structure of non-equilibrium forces. We demonstrate this by presenting exact analytical forms for the optimal protocol and the corresponding work for any driving force and protocol duration. We also derive a general quasistatic bound on the work, relying only on the coarse-grained, time-integrated characteristics of the applied forces. Notably, we show that the optimal protocols automatically harness information about non-equilibrium forces and an initial state measurement to extract work. These findings chart new directions for designing adaptive, energy-efficient strategies in noisy, time-dependent environments, as illustrated through our examples of periodic driving forces and active matter systems. By exploiting the temporal structure of non-equilibrium forces, this largely unexplored approach holds promise for substantial performance gains in microscopic devices operating at the nano- and microscale.

Over two centuries ago, the development of thermodynamics laid the foundation for the Industrial Revolution. In recent decades, major advances, particularly through the development of stochastic thermodynamics, have extended thermodynamic principles to microscopic systems, where thermal fluctuations play a dominant role[1–4]. This emerging framework enables us to rigorously address two central challenges: how to optimally control small-scale processes under constraints of accuracy, speed, and minimal energy expenditure; and how to efficiently harvest energy from strongly fluctuating, far-from-equilibrium environments.

Harvesting energy from non-equilibrium forces and fluctuations has already proven successful in a variety of macroscopic technologies, including wave-energy converters that utilize oscillatory forces[5], piezoelectric devices powered by biomechanical deformation[6–8], and wearables that generate energy from human motion[9]. At microscopic scales, this principle may be even more consequential, as both biological and synthetic systems routinely operate in dynamic, out-of-equilibrium conditions. There is growing evidence that non-

equilibrium fluctuations are not just unavoidable noise, but can be harnessed as a resource. For example, biological systems such as molecular motors perform micro- and nanoscale tasks with remarkable efficiency despite operating under noisy and driven conditions[10,11]. Likewise, non-equilibrium stochastic engines have in some cases been shown to outperform their equilibrium counterparts by exploiting fluctuations[12–17]. A deeper understanding of these effects may prove crucial for the future design of efficient, robust small-scale engines[18].

As technological innovation continues to push the boundaries of miniaturization, identifying the fundamental limits of these processes and designing control strategies that minimize energetic and temporal costs, has become essential for the optimal operation and design of next-generation microscopic machines. Of particular interest is the development of optimal strategies for varying control parameters over time to drive a system between two states while minimizing costs such as energy, dissipation, or duration[19–22]. Recent advances have extended these concepts to more complex, non-homogeneous environments, including disordered media[23,24], stochastic resetting processes[25], and

[1]Institut für Theoretische Physik II - Weiche Materie, Heinrich-Heine-Universität Düsseldorf, Düsseldorf, Germany. [2]Raymond & Beverly Sackler School of Chemistry, Tel Aviv University, Tel Aviv, Israel. [3]Raymond & Beverly Sackler School of Physics & Astronomy, Tel Aviv University, Tel Aviv, Israel. ✉e-mail: kristian.olsen@hhu.de

viscoelastic backgrounds[26]. Optimal protocols have also been studied in systems involving multiple or constrained control parameters[27]. Additionally, approaches based on information geometry and thermodynamic metrics have yielded broad, unifying insights into optimal control in far-from-equilibrium systems[28].

Despite these advances, a critical question remains largely unexplored: how to exploit the temporal structure of forces and fluctuations in non-equilibrium environments. Unlike equilibrium systems, which lack intrinsic time dependence, non-equilibrium settings often exhibit rich temporal features, such as characteristic timescales and frequency spectra, that may be harnessed for improved control and energy extraction. Indeed, generalized Landauer bounds suggest that excess work can be reduced by leveraging the information content separating a system's non-equilibrium state from a reference equilibrium state[29,30].

In this work, we address this critical question by identifying optimal protocols that extract energy using information about time-dependent forces acting on a particle. Since forces induce translational displacements, it is natural to consider protocols that manipulate the trap via a translational degree of freedom, allowing the control strategy to adapt to force-induced effects. These forces may originate from external fields such as fluid flows or electromagnetic fields, or arise internally, as in the case of self-propulsion forces driving active particles (see Fig. 1). Taking a unified perspective, we present a general solution to the optimal control problem that applies broadly to such temporally driven systems. We illustrate this framework through the paradigmatic example of a particle confined in a harmonic potential, where the protocol governs the trap's position.

Considering a protocol $\boldsymbol{\lambda}(t)$ operating in a system subjected to time-dependent forces $\mathcal{F}(t)$, we provide the exact form of the *optimal* protocol for a desired operation: $\boldsymbol{\lambda}(t) = \boldsymbol{\lambda}[t, \mathcal{F}(t)]$. These optimal protocols naturally decompose into a force-independent equilibrium contribution, $\boldsymbol{\lambda}_{\mathrm{eq}}(t)$, and a force-dependent non-equilibrium contribution, $\boldsymbol{\lambda}_{\mathrm{neq}}(t) = \boldsymbol{\lambda}_{\mathrm{neq}}[t, \mathcal{F}(t)]$. We then derive an exact expression for the thermodynamic work, valid for arbitrary driving forces $\mathcal{F}(t)$ and protocol durations, and establish a quasistatic bound on the maximum extractable work. In the slow-driving limit, the total work separates into three distinct contributions: (i) an information-geometric term quantifying how information from an initial non-equilibrium state can be converted into work, (ii) the work required to slowly drag a particle in the presence of time-averaged forces, and (iii) additional work that can be extracted by responding to fast dynamical modes in the driving. We illustrate the general results through several applications, including particles driven by periodic forces and a broad class of time-dependent forces relevant to the field of active matter.

## Results

### Optimal control of a colloidal particle in a harmonic trap far from thermal equilibrium

To illustrate our approach, we examine a well-established model system: a particle confined within a harmonic potential, where the control protocol governs the position of the trap center. An object, under the combined effect of a smooth external driving force $\mathbf{f}(t)$, a harmonic potential with stiffness $k$ and trap center $\boldsymbol{\lambda}(t)$, and a thermal bath at temperature $T$, obeys the Langevin equation

$$\gamma\dot{\mathbf{x}}(t) = -k[\mathbf{x}(t) - \boldsymbol{\lambda}(t)] + \mathbf{f}(t) + \sqrt{2k_B T\gamma}\,\boldsymbol{\xi}(t), \tag{1}$$

where $\boldsymbol{\xi}(t)$ is Gaussian white noise with $\langle\boldsymbol{\xi}(t)\rangle = 0$ and $\langle\boldsymbol{\xi}(t)\boldsymbol{\xi}(t')\rangle = \delta(t - t')$. The bath is unaffected by the forces $\mathbf{f}(t)$[31]. This model could represent a colloidal particle manipulated by optical tweezers, or serve as a toy model for a motion protocol $\boldsymbol{\lambda}(t)$ for a micro-bot with spring-like coupling to a cargo at position $\mathbf{x}(t)$. The forces $\mathbf{f}(t)$, which for now are arbitrary and may be deterministic or stochastic, generically drive the system out of equilibrium. We emphasize that it is assumed that these driving forces are spatially-homogeneous, as other scenarios would render the problem non-linear and analytically untractable. Yet, a wide range of non-equilibrium systems can be captured by the above model, as we describe in future sections. We denote by $\mathbf{q}(t) \equiv \langle\mathbf{x}(t)\rangle$ the mean particle trajectory, where the average is taken with respect to the Gaussian noise as well as any possible stochastic effects in the forces. We also let $\mathcal{F}(t) \equiv \langle\mathbf{f}(t)\rangle$ denote the averaged force. The mean particle trajectory can be obtained by averaging the equation of motion, yielding

$$\gamma\dot{\mathbf{q}}(t) = -k[\mathbf{q}(t) - \boldsymbol{\lambda}(t)] + \mathcal{F}(t). \tag{2}$$

We emphasize that if the forces are deterministic and known with precision, $\mathcal{F}(t) = \mathbf{f}(t)$. Furthermore, any stochastic effect may be either inherent to the forces, such as in the case of active self-propulsion forces, or represent effective forces resulting from errors in experimental measurements or inference. Recent efforts in force inference have employed a variety of approaches, including machine learning, Bayesian methods, and information-theoretic techniques[32–39]. This underscores the importance of *information-limited optimization*, where protocols are derived based on the *perceived* driving forces-

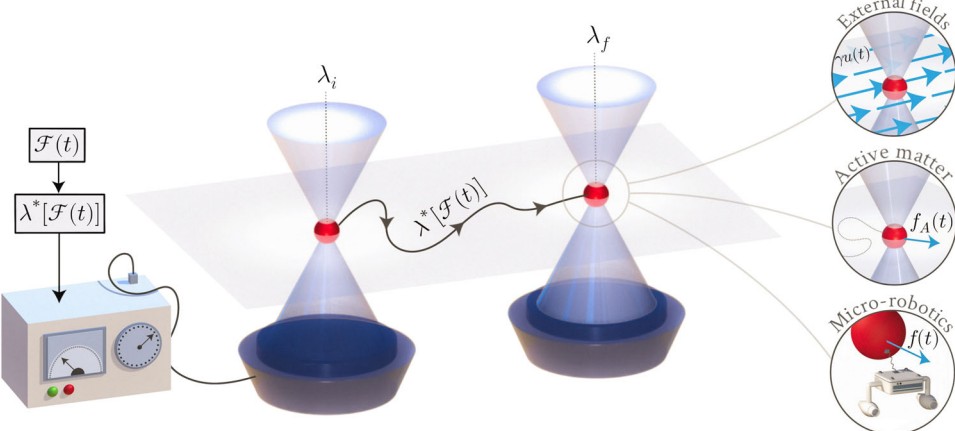

**Fig. 1 | Optimal protocols for particle transport are derived for particles under the influence of arbitrary time-dependent forces $\mathcal{F}(t)$.** Given knowledge about these forces, a harmonic trap with a movable center $\boldsymbol{\lambda}(t)$ is tuned by the controller, at the cost of thermodynamic work. The forces may represent external drives such as applied fields or flows, or internally generated forces such as in active matter. The optimal transport problem can also be interpreted in terms of micro-robotics and cargo transport, where $\boldsymbol{\lambda}^{*}(t)$ represents the trajectory of a micro-robot transporting a cargo that is exposed to non-equilibrium conditions.

those accessible through coarse-graining, averaging, or measurement uncertainty. For instance, if the true forces $\mathbf{f}(t; \kappa)$ depend on a parameter $\kappa$ known only with finite precision, typically distributed according to a Gaussian prior, it is natural to work with effective forces $\mathcal{F}(t) = \langle \mathbf{f}(t; \kappa) \rangle_\kappa$ averaged over the distribution of measured values. These effective forces $\mathcal{F}(t)$, appearing in the averaged dynamics, reflect the information available to the controller and represent the reproducible driving conditions across repeated experiments or simulations.

Controlling the position of the harmonic trap $V[\mathbf{x}, \boldsymbol{\lambda}(t)] = \frac{1}{2}k[\mathbf{x} - \boldsymbol{\lambda}(t)]^2$ with a protocol $\boldsymbol{\lambda}(t)$ comes at a thermodynamic cost, which is given by the mean work

$$\mathcal{W}[\{\boldsymbol{\lambda}(t)\}_0^{t_f}] = \int_0^{t_f} dt \left\langle \dot{\boldsymbol{\lambda}}(t) \cdot \frac{\partial V[\mathbf{x}(t), \boldsymbol{\lambda}(t)]}{\partial \boldsymbol{\lambda}(t)} \right\rangle. \quad (3)$$

We use the convention that a positive work implies energy has to be paid, while negative work corresponds to energy extracted. In Eq. (3), averages are taken over noise and stochastic force realizations as well as over measurement errors. Under the boundary conditions $\boldsymbol{\lambda}(t_0) = \boldsymbol{\lambda}_i$ and to $\boldsymbol{\lambda}(t_f) = \boldsymbol{\lambda}_f$, we seek the optimal protocol $\boldsymbol{\lambda}^*(t)$ that minimizes the work performed over a fixed time interval. In the absence of external forces ($\mathcal{F} = 0$), the optimal protocol is known to be linear, with symmetric discontinuous jumps at the very beginning and end of the protocol[19]. Using Eq. (2) as a dynamical constraint together with the work in Eq. (3), the optimization problem can be solved exactly, resulting in an Euler–Lagrange equation $\gamma \ddot{\mathbf{q}}(t) = \dot{\mathcal{F}}(t)/2$. This can be solved with the aforementioned boundary conditions, from which both the optimal protocol, mean particle position, and work can be calculated exactly. See the "Methods" section for technical details.

Under general $\mathcal{F}(t)$, we derive an optimal protocol which can be decomposed into two contributions

$$\boldsymbol{\lambda}_*(t) = \boldsymbol{\lambda}_{eq}(t) + \boldsymbol{\lambda}_{neq}(t). \quad (4)$$

The first term is an equilibrium contribution ($\mathcal{F} = 0$), and the second a non-equilibrium contribution ($\mathcal{F} \neq 0$) that is determined solely by the perceived drive and protocol duration. Respectively, these take the form

$$\boldsymbol{\lambda}_{eq}(t) = \mathbf{q}_i + \frac{1 + \omega t}{2 + \omega t_f}(\boldsymbol{\lambda}_f - \mathbf{q}_i), \quad (5)$$

$$\boldsymbol{\lambda}_{neq}(t) = \int_0^t dt' \frac{\mathcal{F}(t')}{2\gamma} - \frac{1 + \omega t}{2 + \omega t_f} \int_0^{t_f} dt' \frac{\mathcal{F}(t')}{2\gamma} - \frac{\mathcal{F}(t)}{2\gamma\omega}, \quad (6)$$

where $\omega = k/\gamma$ is the inverse relaxation timescale of the harmonic trap. In the free diffusive limit, $\mathcal{F} = 0$ we recover the protocol first obtained in Ref. 19 in the case of an initial equilibrium state with $\mathbf{q}_i = 0$. In this part of the protocol, $\boldsymbol{\lambda}_{eq}(t)$, consists of a straight line as a function of time but with discontinuous jumps at the beginning and end of the protocol. The non-equilibrium contribution to the protocol $\boldsymbol{\lambda}_{neq}(t)$, containing the force, depends only on the duration of the protocol, but not on the initial and final location of the protocol. Hence, Eq. (4) can be interpreted as the equilibrium protocol with superimposed corrections that compensate for the driving forces $\mathcal{F}(t)$. During the protocol, the mean particle path is given by

$$\mathbf{q}(t) = \mathbf{q}_i + \left[\frac{\boldsymbol{\lambda}_f - \mathbf{q}_i}{2 + \omega t_f} - \frac{1}{2 + \omega t_f} \int_0^{t_f} dt' \frac{\mathcal{F}(t')}{2\gamma}\right] \omega t + \frac{1}{2\gamma} \int_0^t dt' \mathcal{F}(t'). \quad (7)$$

We note that there are two contributions to the path; first, a linear contribution that depends on both the details of the dynamics and protocol, and a second, potentially non-linear time-dependence coming from the last term of Eq. (7).

We emphasize that, just like in ref. 19, the protocol only ensures that the potential is at the final location $\boldsymbol{\lambda}_f$ at time $t_f$, without any constraints on particle location. One could in principle, also constrain the particle location at the end of the protocol, leading to increased control, but at the cost of less energy extraction. In this relation between control and cost, we consider protocols that are able to extract maximal work from the non-equilibrium forces. The particle position at the end of the protocol $\mathbf{q}(t_f)$ will, in the quasistatic regime, be given as

$$\lim_{t_f \to \infty} \mathbf{q}(t_f) = \boldsymbol{\lambda}_f + \lim_{t_f \to \infty} \frac{\overline{\mathcal{F}}(t_f)}{k}, \quad (8)$$

where $\overline{\mathcal{F}}(t_f)$ is the time-averaged force (see Eq. (14) in the following). Hence, the final particle position will, even in the quasistatic regime, deviate from the target location, in contrast to equilibrium systems[19]. Surprisingly, this deviation is not determined by the value of the force at the later stages of the protocol, but by the full time-averaged force since the initial time $t = 0$. The memory of the full dynamics is a consequence of the way in which the optimization intertwines the forces, the protocol, and the particle position.

As shown in the "Methods" section, the work associated with the optimal protocol can be shown to take the form

$$\mathcal{W} = \frac{1}{2}k(\boldsymbol{\lambda}_f - \mathbf{q}(t_f))^2 - \frac{1}{2}k(\boldsymbol{\lambda}_i - \mathbf{q}_i)^2$$
$$+ \left(\omega \frac{\boldsymbol{\lambda}_f - \mathbf{q}_i}{2 + \omega t_f} - \frac{\omega}{2 + \omega t_f} \int_0^{t_f} dt \frac{\mathcal{F}}{2\gamma}\right)^2 \gamma t_f \quad (9)$$
$$- \frac{1}{\gamma} \int_0^{t_f} dt \left(\frac{\mathcal{F}}{2}\right)^2$$

which is an exact result valid for arbitrary protocol durations $t_f$ and for arbitrary forces $\mathcal{F}(t)$.

## Quasistatic bound on work extraction

In many cases, knowing the quasistatic limit is informative, as it provides bounds on the work exchanged. Whether this bound is positive (costing work) or negative (extracting work) and bounded or infinite is of high practical relevance.

Here, the quasistatic limit of Eq. (9), $\mathcal{W}_{qs} = \lim_{t_f \to \infty} \mathcal{W}$, is useful in several ways. Firstly, it offers intuition behind the terms contributing to the work and provides physical insights into the mechanisms by which work is extracted from knowledge of the forces. Secondly, the quasistatic limit may be relevant to slow but finite-time experiments. Since, for an optically trapped particle, the potential does not change shape during the protocol, the equilibrium free energy difference is zero. Consequently, in the quasistatic limit, only non-equilibrium effects contribute, arising either from non-equilibrium initial conditions captured by $\mathbf{q}_i$ or from non-equilibrium driving forces $\mathcal{F}(t)$. Taking the slow limit ($t_f \to \infty$) of Eq. (9), we find

$$\mathcal{W}_{qs} = \mathcal{W}_i + \mathcal{W}_{ta} + \mathcal{W}_d, \quad (10)$$

where the work has been decomposed into three parts; an information theoretic contribution, a contribution from time-averaged forces, and a contribution from forces deviating from its time-average, respectively taking the form

$$\mathcal{W}_i = -\frac{1}{2}k(\boldsymbol{\lambda}_i - \mathbf{q}_i)^2, \quad (11)$$

$$\mathcal{W}_{ta} = -(\mathbf{q}_f - \mathbf{q}_i) \lim_{t_f \to \infty} \overline{\mathcal{F}}(t_f), \quad (12)$$

$$\mathcal{W}_d = -\frac{1}{4\gamma} \lim_{t_f \to \infty} t_f \mathrm{Var}(\mathcal{F}; t_f). \quad (13)$$

In these expressions, we also used the time-averaged mean and variance

$$\overline{\mathcal{F}}(t_f) = \frac{1}{t_f} \int_0^{t_f} dt' \mathcal{F}(t'), \tag{14}$$

$$\mathrm{Var}(\mathcal{F}; t_f) = \frac{1}{t_f} \int_0^{t_f} dt' \mathcal{F}^2(t') - \left[\frac{1}{t_f} \int_0^{t_f} dt' \mathcal{F}(t')\right]^2. \tag{15}$$

Before we interpret each term in the quasistatic work, it is worth emphasizing the simplicity of this result. All terms in Eq. (10) may be calculated directly through the forces acting on the free particle $\mathcal{F}(t)$ in addition to the fixed boundary conditions of the protocol $\{\boldsymbol{\lambda}_i, \boldsymbol{\lambda}_f\}$. Hence, this formula can be applied to a wide range of systems without having to go through the optimization procedure explicitly. Furthermore, in the quasistatic limit, the work is determined solely by the first two time-integrated cumulants, rendering higher-order fluctuations irrelevant.

In Eq. (10), the first term is determined by the initial condition of the particle and naturally admits an information-theoretic interpretation. Because we specify only the mean of the initial distribution $p_i(\mathbf{x})$, the system may start in an arbitrarily complex state. However, the harmonic trap can only access the portion of this information that is compatible with its fixed shape or position[40,41].

To make this more precise, we introduce the M-projection $\pi[p_i]$, defined by ref. 42 as

$$\pi[p_i](\mathbf{x}) = \mathrm{argmin}_{\rho \in \mathcal{B}} D_{KL}(p_i \parallel \rho), \tag{16}$$

where $D_{KL}$ is the Kullback–Leibler divergence. This operation projects the initial distribution $p_i(\mathbf{x})$ onto the space $\mathcal{B} = \{Z^{-1}e^{-\beta V[\mathbf{x}, \boldsymbol{\mu}]}, \mu \in \mathbb{R}\}$ of Boltzmann states compatible with the trap manipulation-here, fixed-variance (i.e., constant $\beta$ and stiffness) Gaussian densities with variable location $\mu$ (often called *shift measures*). By minimizing $D_{KL}$, $\pi[p_i]$ is the least-information-loss Gaussian approximation of $p_i$.

In our example, $\pi[p_i](\mathbf{x})$ becomes a Gaussian with center $\mathbf{q}_i$ and variance determined by the trap shape. One can then show that

$$k_B T D_{KL}(\pi[p_i] \parallel p_{eq}) = \frac{1}{2}k(\boldsymbol{\lambda}_i - \mathbf{q}_i)^2 \tag{17}$$

where $p_{eq}$ is the true Boltzmann state of the initial trap. Thus, the first term in Eq. (10) quantifies the *accessible* information within the non-equilibrium initial state that can be transformed into negative, i.e., extracted, work. A detailed derivation of this can be found in the Supplementary Information.

The two last terms of Eq. (10) are contributions originating in the non-equilibrium driving forces. More precisely, the contribution $\mathcal{W}_{ta}$ originates in the time-averaged force and is simply the work needed to move a particle a distance $\mathbf{q}_f - \mathbf{q}_i$ in the presence of $\overline{\mathcal{F}}$. The last term is determined by the deviations of the force around the time-averaged mean, and can be written

$$\mathcal{W}_d = -\frac{1}{4\gamma} \lim_{t_f \to \infty} \int_0^{t_f} d\tau \boldsymbol{\delta}\mathcal{F}(\tau)^2 \tag{18}$$

where $\boldsymbol{\delta}\mathcal{F}(\tau) = \mathcal{F}(\tau) - \overline{\mathcal{F}}(t_f)$ are the deviations of the force from its time averaged value. This term encodes how much work can be extracted by utilizing the information about the temporal details of the force. We emphasize here that one can extract more work from time-varying forces than from stationary ones, especially in situations where the deviations from the time-averaged mean are strongly persistent in time.

To gain further insight into this contribution, $\mathcal{W}_d$, consider a key outcome of the optimization procedure: the Euler–Lagrange equation implies an effective overdamped motion for the particle's mean position. In the quasistatic limit, this takes the simple form $\gamma \dot{\mathbf{q}}(t) = \boldsymbol{\delta}\mathcal{F}(t)/2$. As discussed in the Supplementary Information, this effective equation describes a *free* particle driven by the forces $\boldsymbol{\delta}\mathcal{F}(t)/2$. One may interpret this as capturing the fast velocity modes of the particle; meanwhile, the slow velocity mode, which transports the particle a finite distance over an infinite time vanishes in the quasistatic limit, and its associated work is accounted for by $\mathcal{W}_{ta}$.

Because the particle effectively experiences the force $\boldsymbol{\delta}\mathcal{F}(t)/2$, there is a corresponding work contribution with differential increment $dW_{\boldsymbol{\delta}\mathcal{F}} = \frac{\boldsymbol{\delta}\mathcal{F}}{2}d\mathbf{q}(t)$. From the effective equation of motion, $d\mathbf{q}(t) = \frac{\boldsymbol{\delta}\mathcal{F}}{2\gamma}dt$, so the instantaneous power becomes $\dot{W}_{\boldsymbol{\delta}\mathcal{F}} = \frac{\boldsymbol{\delta}\mathcal{F}^2}{4\gamma}$. Integrating over the duration of the protocol yields the work done by these forces, which is precisely the extractable work $\mathcal{W}_d = -\int_0^{t_f} dt \dot{W}_{\boldsymbol{\delta}\mathcal{F}}(t)$. All contributions to the work, Eq. (10), are summarized in Fig. 2.

## Precision vs. risk in erroneous force inference

Here we consider the errors occurring if one rather than using the true force $\mathcal{F}$ uses a coarse-grained version $\tilde{\mathcal{F}}$ coming for example, from errors in inference. Since exact inference is hard or impossible, knowledge about what determines the error in the estimated work extraction could be crucial in realistic implementations. Here we derive a simple bound on this error, expressed fully in terms of experimentally accessible quantities.

In the following, we will by $\boldsymbol{\lambda}(t)$ always mean the protocol derived using the inferred or estimated forces $\tilde{\mathcal{F}}$. Let $\mathbf{q}(t)$ and $\tilde{\mathbf{q}}(t)$ respectively denote the mean position under the effect of the true force (actual mean trajectory) and the expected trajectory under the belief that the forces are $\tilde{\mathcal{F}}(t)$. The work is linear in the position, so we can write

$$\Delta\mathcal{W} \equiv \mathcal{W}[\mathbf{q}] - \mathcal{W}[\tilde{\mathbf{q}}] = \int_0^{t_f} dt \dot{\boldsymbol{\lambda}}(t)[\tilde{\mathbf{q}}(t) - \mathbf{q}(t)]. \tag{19}$$

The particle position takes the general form

$$\mathbf{q} = \mathbf{q}_0 e^{-\omega t} + \int_0^t ds e^{-\omega(t-s)}[\gamma^{-1}\mathcal{F}(s) + \omega\boldsymbol{\lambda}(s)] \tag{20}$$

Using this expression in the work error above, we find

$$\Delta\mathcal{W} = \omega \int_0^{t_f} dt \dot{\boldsymbol{\lambda}}(t) \int_0^t ds e^{-\omega(t-s)}[\tilde{\mathcal{F}}(s) - \mathcal{F}(s)] \tag{21}$$

If one knew the true forces $\mathcal{F}$, there would be no issue with inference. Hence, often one can at best know $|\tilde{\mathcal{F}}(s) - \mathcal{F}(s)| \le \varepsilon$, with $\varepsilon$ a uniform bound on the inference errors. Hence

$$|\Delta\mathcal{W}| \le \omega\varepsilon \int_0^{t_f} dt |\dot{\boldsymbol{\lambda}}(t)| \int_0^t ds e^{-\omega(t-s)} \tag{22}$$

$$= \varepsilon \int_0^{t_f} dt |\dot{\boldsymbol{\lambda}}(t)|(1 - e^{-\omega t}) \tag{23}$$

$$\le \varepsilon \int_0^{t_f} dt |\dot{\boldsymbol{\lambda}}(t)| \tag{24}$$

where we used monotonicity of integrals. Hence, we have the error bound

$$|\Delta\mathcal{W}| \le \varepsilon \int_0^{t_f} dt |\dot{\boldsymbol{\lambda}}(t)| \tag{25}$$

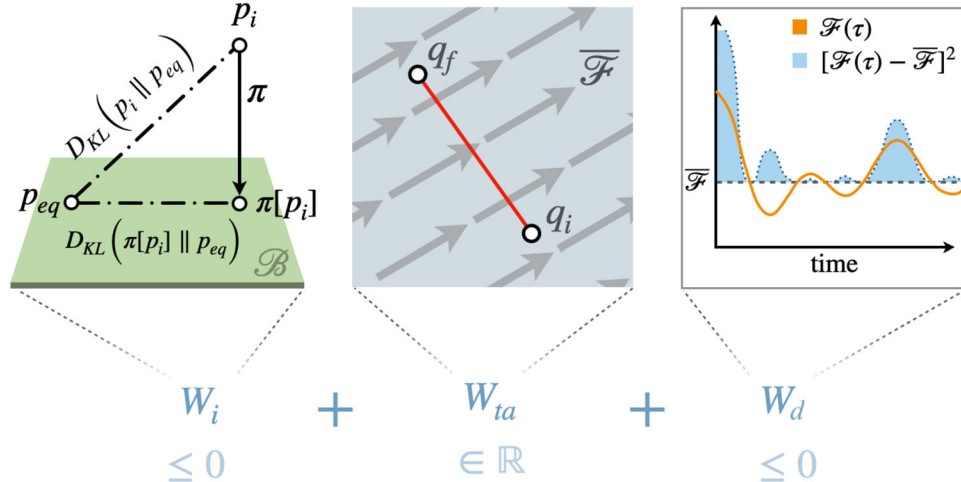

**Fig. 2 | Contributions to the work in the quasistatic limit; a information theoretic work $\mathcal{W}_i$, work due to time-averaged forces $\mathcal{W}_{ta}$, and work due to force deviations from its time-average $\mathcal{W}_d$.** The information theoretic contribution measures the accessible information in the initial non-equilibrium state, which can be extracted as work. The contribution from the time-averaged force can take any sign depending on the direction of forces and desired translation. Finally, deviations around the time-averaged force can be used to extract work, by an amount proportional to the shaded blue area (accumulated square deviations).

The above integral is sometimes called the total variation $\mathcal{TV}[\boldsymbol{\lambda}] = \int_0^{t_f} dt |\dot{\boldsymbol{\lambda}}(t)|$, and hence the work error is bounded by the protocol's total variation times the force inference errors $|\Delta\mathcal{W}| \leq \varepsilon \mathcal{TV}[\boldsymbol{\lambda}]$. The linearity in the force errors is a direct consequence of the linearity of the problem in the presence of harmonic traps, and we would expect non-linear dependence on force errors in other scenarios.

This result tells us that more erratic control protocols with a large total variation have a higher potential for producing errors. The error is also additive in the protocol duration; increasing $t_f$ gives more chances to make unfavorable moves based on faulty knowledge. This insight could also aid in choosing the type of inference one should use. Indeed, if multiple methods are available, $\boldsymbol{\lambda}$ and thereafter $\mathcal{TV}[\boldsymbol{\lambda}]$ should be calculated in each scenario, and the one with smallest total variation should be applied. This analysis demonstrates an inherent tradeoff between precision and gain: more erratic control protocols can, in principle, exploit fine-scale details of the forces to extract additional work, but they also carry a higher risk of error. Conversely, smoother protocols with lower total variation reduce the likelihood of mistakes, at the potential cost of leaving some work unextracted. Understanding this balance allows one to tailor control strategies according to whether the priority is maximizing work extraction or minimizing errors, effectively framing the problem as a risk-reward optimization.

## Case studies
Our above framework leverages information about non-equilibrium forces to enhance energetic efficiency. This is not unlike information engines, which traditionally utilize information through measurement and feedback to rectify fluctuations and extract work[30,43,44]. Leveraging prior knowledge of the forces that drive the system away from equilibrium, Euler–Lagrange minimization gives rise to protocols that both anticipate and adapt to prescribed particle dynamics, enabling the spontaneous extraction of maximal energy from non-equilibrium processes. In the Supplementary Information, a baseline scenario without forces is discussed. Below, we explore two case studies that illustrate the versatility of our approach with respect to different force types: externally applied periodic forces and internally generated active self-propulsion forces.

## Case I: periodic forces
Many energy harvesting solutions are based on periodic forces or motion, such as wave-energy converters, wearable fabrics that extract energy from movement, and piezoelectric generators that can charge pacemakers through heartbeats[5,6,9,45,46]. These classical approaches typically utilize mechanical oscillators coupled to transducers, aiming to convert displacement, velocity, or strain into electrical power. Here, we consider a simple microscopic analogy, which in contrast relies on optimal control to identify limits and optimal strategies for energy harvesting, independent of any specific transducer design. We consider a Brownian particle effectively confined to one-dimensional movement, exposed to periodic driving forces

$$\mathcal{F}(t) = f_0 \sin(t/\tau_p). \tag{26}$$

Here $f_0$ is the amplitude of the force while $\tau_p$ determines its periodicity. Figure 3 shows the work as a function of protocol duration and force periodicity. The associated optimal protocol is shown in regions of both positive and negative work. As is common for optimal protocols, discontinuous jumps are seen in the beginning and final parts of the protocol.

We see that for sufficiently slow protocols, compared to the forcing period, the optimal control is able to utilize the oscillations such that work can be extracted. The optimal protocol harnesses the dynamic information available and automatically extracts as much work as possible. This is a consequence of precise knowledge of the force at all times. The protocol consists of repeatedly letting the force move the particle into a high-energy state before shifting the potential accordingly to extract the stored energy as work. Let us recall that in Brownian information engines, feedback is used to rectify thermal noise and convert measurement information into work. In simple engine designs, the potential has the freedom to move left and right based on the measurement outcome, extracting work is a manner similar to the above protocols[47]. Figure 4 summarizes the main mechanism behind the work extraction in the case of periodic forces.

While the above repeating cycles are specific to the current example, the way in which work is extracted offers insights into more generic situations. Indeed, combining the Euler–Lagrange equation with the equation of motion, we have $\dot{\boldsymbol{\lambda}} = \dot{\mathbf{q}} - \dot{\mathcal{F}}/(2k)$. When forces increase in a given direction $\dot{\mathcal{F}} > 0$ the optimal protocol lets the particle move faster than the trap, $\dot{\boldsymbol{\lambda}} < \dot{\mathbf{q}}$, lifting the particle to a higher energy state. Once forces start to decrease $\dot{\mathcal{F}} < 0$ the protocol catches up to the particle $\dot{\boldsymbol{\lambda}} > \dot{\mathbf{q}}$.

The above cycle can, in principle, extract arbitrarily large amounts of work over an indefinitely long protocol. In our example, this follows from Eq. (10), where $\mathrm{Var}(\mathcal{F}) = f_0^2/2$ remains constant, causing $\mathcal{W}_d \sim$

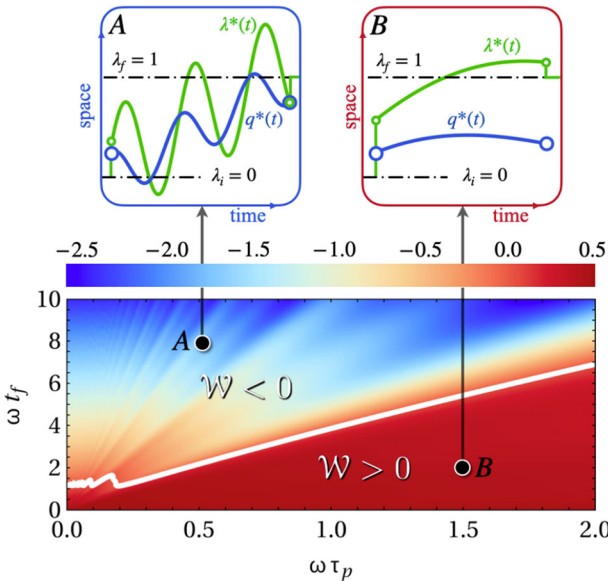

**Fig. 3 | Work as a function of the timescale of force oscillations $\tau_p$ and protocol duration $t_f$, showing regions where work must be paid or can be extracted.** Insets (points A and B) show optimal protocol (green line) and the associated mean particle trajectory (blue line). Parameters used are $\omega = \gamma = \lambda_f = 1, f_0 = -1, \lambda_i = 0, q_i = \lambda_f/4$.

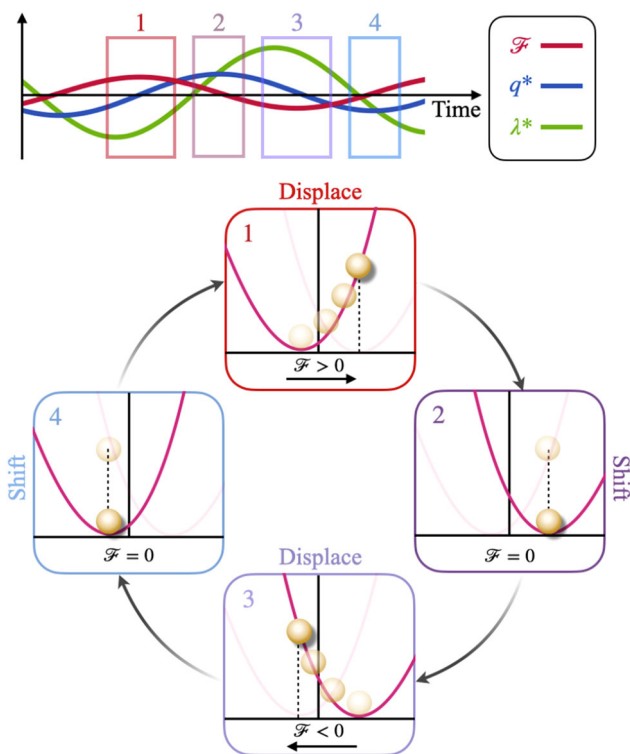

**Fig. 4 | Sketch of the mechanism behind work extraction for periodic forcing. Top panel shows forces (red line), particle position (blue line), and protocol (green line) as a function of time.** When the forces are approximately maximal, the particle climbs the potential (panel 1). Once the maximal position is reached, the protocol switches from negative to positive side and catches up to the particle (panel 2). In panels 3 & 4, this process repeats in the opposite direction.

$t_f \mathrm{Var}(\mathcal{F})$ to grow unbounded (in the negative direction) as $t_f \to \infty$. For finite yet large protocol durations, $t_f \gg \tau_p$, the term $\mathcal{W}_d$ dominates, and the work behaves as

$$\mathcal{W} = -\frac{f_0^2}{8\gamma} t_f \qquad (27)$$

which is independent of the forcing period $\tau_p$. This results from a direct competition between two effects caused by changing $\tau_p$; firstly, it affects the amplitude of the particle displacements and hence how much energy could be harvested per cycle. Secondly, it also changes how many cycles of harvesting can be completed in a given total duration $t_f$. For slow protocols, these effects balance, resulting in Eq. (27).

This large potential for work extraction relies on precise initial information about the periodic forces. For example, we can easily extend the model by considering a force $f(t) = f_0 \sin(t/\tau_p + \varepsilon)$ where $\varepsilon$ is a random variable representing our ignorance about phase information. Taking these errors to be normal distributed with zero mean and variance $\sigma_\varepsilon^2$, we obtain the mean force $\mathcal{F}(t) = \exp(-\sigma_\varepsilon^2/2) f_0 \sin(t/\tau_p)$. Hence, incorporating these initial errors effectively rescales the force amplitude, and the work extracted at large protocol durations instead takes the form $\mathcal{W} = -\frac{f_0^2 \exp(-\sigma_\varepsilon^2)}{8\gamma} t_f$. Hence, initial measurement errors can be detrimental to the design of engines, leading to exponentially reduced work extraction. Notably, while phase errors cause an exponential suppression of work, their degree of suppression remains unchanged over time. Once the phase is incorrectly estimated, the mismatch persists, effectively lowering the force amplitude throughout the entire process.

In the presence of a periodicity error, modeled as $f(t) = f_0 \sin(t/(\tau + \varepsilon))$ the effective force can be approximated for small $\varepsilon$. When $\varepsilon$ is normally distributed with a small variance $\sigma_\varepsilon$ we use:

$$\mathcal{F}(t) \approx f_0 \left\langle \sin\left(\frac{t}{\tau_p} - \frac{t}{\tau_p^2}\varepsilon\right)\right\rangle \qquad (28)$$

$$= f_0 \exp\left(-\frac{t^2\sigma_\varepsilon^2}{2\tau_p^4}\right) \sin\left(\frac{t}{\tau_p}\right). \qquad (29)$$

Here, the periodic force experiences a pronounced exponential suppression. In contrast to the case where the error is in the phase, the error in periodicity gives rise to a larger phase mismatch over time, leading to strong destructive interference, which suppresses the effective forces. The work in the quasistatic case can be calculated as before,

$$\mathcal{W}_{qs} \approx -\frac{\sqrt{\pi}\tau_p^2 f_0^2}{16\gamma\sigma_\varepsilon}\left(1 - e^{-\tau_p^2/\sigma_\varepsilon^2}\right). \qquad (30)$$

In sharp contrast to the case without errors, work extraction is now bounded. We emphasize that although this approximation may not be quantitatively exact in the quasistatic approximation, it clearly shows how small errors qualitatively affect the extracted work. It is also worth emphasizing that when there are errors in the periodicity, the quasistatic work extraction becomes smaller when the periodicity $\tau_p$ is reduced. This reflects the fact that for very rapid forces, even small errors would make work extraction hard.

## Case II: Work extraction from active forces

Active matter represents a compelling category of non-equilibrium systems, where mechanisms for work extraction have been investigated recently[48–58]. Unlike the externally imposed oscillatory forces discussed previously, active particles generate propulsive forces internally through autonomous mechanisms. We now apply our general framework to several representative active matter scenarios and compare the resulting work that can be extracted in each case.

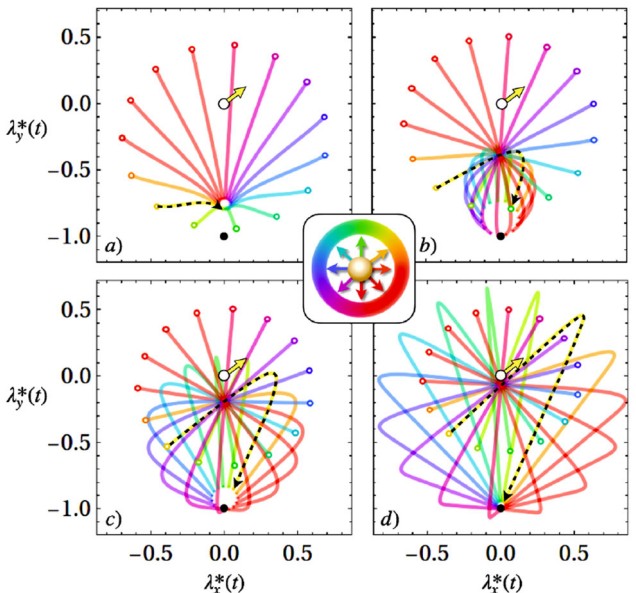

**Fig. 5 | Optimal protocols for an active Brownian particle (ABP) starting at the origin with various initial orientations $\phi_0$ indicated by the color coding in the central inset.** Each trajectory shows the trap location in the $x-y$ plane, in units of the persistence length $\ell_p = f_0/(\gamma D_r)$. The protocol begins at the origin, where the particle is confined in a harmonic potential (empty black circle), and ends at $(0, -1)$ (filled black circle). **a–d** show results for $D_r t_f = 2, 5, 10$, and $30$, respectively. In all cases, the protocol features an initial discontinuous jump from the empty black circle to the small empty colored circle, as well as a jump at the end of the protocol. The dashed black line highlights the protocol corresponding to the initial orientation indicated by the yellow arrow, including the initial and final instantaneous jump. Parameters: $D_r\tau_R = 2, f_0 = \omega = \gamma = 1$.

We consider an active particle in two dimensions, modeled as an active Brownian particle (ABP), obeying the stochastic equation of motion

$$\dot{\mathbf{x}}(t) = \frac{f_0}{\gamma}\hat{\mathbf{n}}(t) - \omega[\mathbf{x}(t) - \boldsymbol{\lambda}(t)] + \sqrt{2D}\boldsymbol{\xi}(t) \qquad (31)$$

where $f_0$ is the constant magnitude of the active self-propulsion force, and $\hat{\mathbf{n}}(t) = [\cos\phi(t), \sin\phi(t)]$ determines the direction of propulsion. Persistence is expected to play a key role in the energy extraction in any active system, motivating us to consider not only reorientations driven by white noise, as is usual for ABPs, but a slightly more realistic situation where finite-time correlations are present. In the simplest case, one can consider orientations driven by an Ornstein-Uhlenbeck noise

$$\dot{\phi}(t) = \Omega(t) \qquad (32)$$

$$\dot{\Omega}(t) = -\frac{1}{\tau_R}\Omega + \frac{\sqrt{2D_r}}{\tau_R}\eta(t), \qquad (33)$$

where $D_r$ is a rotational diffusion coefficient and $\tau_R$ a relaxation time-scale associated with the rotational dynamics. Such models have been considered in the past to include memory effects in the orientational dynamics, resulting for example from misalignments with the instantaneous propulsion direction or inertial effects[59–62]. When $\Omega$ is stationary, the distribution of $\phi(t)$ is known to be Gaussian with mean $\phi_0$ and variance $\sigma_\phi^2(t) = 2D_r(t - \tau_R(1 - e^{-t/\tau_R}))$. This results in a mean force

$$\mathcal{F}(t|\phi_0) = \langle f_0\hat{\mathbf{n}}(t)|\phi_0\rangle = f_0\begin{bmatrix} \cos(\phi_0)e^{-\frac{1}{2}\sigma_\phi^2(t)} \\ \sin(\phi_0)e^{-\frac{1}{2}\sigma_\phi^2(t)} \end{bmatrix}, \qquad (34)$$

conditioned on knowing the initial force, i.e., propulsion direction, as can be obtained from a measurement. From this, the optimal protocol and the associated work can be calculated. Recently, repeating this scheme was proposed as a way of constructing active information engines[55,56]. In the case of one-dimensional run-and-tumble or active Ornstein-Uhlenbeck particles, we recover the results of ref. 55, while our general results can also be used to study a wide range of other active matter models, also in higher dimensions.

Determining the optimal protocol and its work cost requires the time-integrated mean and variance of the force, which we compute exactly (see Supplementary Information). With the time-integrated mean and variance in hand, we compute the optimal protocol $\boldsymbol{\lambda}^*(t)$ directly from Eqs. (5) and (6), with the results shown in Fig. 5. We see that short protocol durations (e.g., panel a) results in protocols with large initial jump and almost linear dragging. Longer protocol durations, however, (e.g., panel d) shows smaller jumps and a curved protocol trajectory that utilizes the particle persistence to lower the energetic cost, or even allow work extraction. Generally, the protocol makes a jump to a position behind the particle, and follows the particle's initial direction of motion for a while. This slows the particle down and after the particle's persistence is lost due to rotational noise, the protocol drags the particle back to the target location.

In analyzing work extraction, the quasistatic bound $\mathcal{W}_{qs}$ can be directly obtained from Eq. (10). Without loss of generality, we set $\phi_0 = 0$, which, by symmetry, implies $\mathcal{F}_y = \lambda_y^*(t) = 0$. As a result, only the $x$-component of the protocol contributes and needs to be considered. This results in

$$\frac{\mathcal{W}_{qs}(\mathfrak{D}_0)}{\mathcal{W}_{qs}(0)} = \frac{e^{2\mathfrak{D}_0}}{(2\mathfrak{D}_0)^{2\mathfrak{D}_0-1}}\left[\Gamma(2\mathfrak{D}_0) - \Gamma(2\mathfrak{D}_0|2\mathfrak{D}_0)\right] \qquad (35)$$

which is expressed solely in terms of the dimensionless number $\mathfrak{D}_0 = D_r\tau_R$, introduced as a *delay number* in Ref. 61. Here, we normalized the work by its $\tau_R \to 0$ value

$$\mathcal{W}_{qs}(0) \equiv \mathcal{W}_{qs}^{ABP} = -f_0^2/(8D_r\gamma), \qquad (36)$$

which is the bound for the normal ABP model with Gaussian white noise driving the orientations. Figure 6, left panel, shows the optimal protocol and the associated mean particle position for a persistent active particle initially oriented along the positive $x$-axis. The optimal protocol exhibits an initial jump behind the particle position. Although this incurs a high initial work cost (as also reported in ref. 55), it strategically enables work extraction from the particle's persistence throughout the remainder of the protocol. The right panel of Fig. 6 shows that the work in the quasistatic limit increases monotonically with the delay number $\mathfrak{D}_0$, confirming the beneficial effect of finite-time angular correlations. We emphasize that while the ratio $\mathcal{W}_{qs}(\mathfrak{D}_0)/\mathcal{W}_{qs}(0)$ is positive, the work $\mathcal{W}_{qs}(\mathfrak{D}_0)$ remains strictly negative across all delay number values, indicating effective work extraction from the system.

The quasistatic work bound given by Eq. (10) can easily be evaluated for a wide range of other active particle models as well. In Table 1 we list four other active particle models and the quasistatic work associated with their optimal protocols. More precisely, we consider active particles driven by fractional Brownian noise[63], a random rotational diffusion model, chiral particles[64,65], and particles that accelerate or decelerate in time[66]. For the random rotational diffusivity model with mean fixed to the rotational diffusion of a pure ABP, we find double the potential for work extraction, while chirality does not affect the bound. For fractional angular noise and accelerating ABPs, the work extraction may be higher or lower than the pure ABP case, depending on parameters.

## Discussion

As technology continues to advance toward miniaturization, realistic microscale devices will increasingly operate in complex, dynamic environments shaped by time-dependent forces and fluctuations. In such settings, optimal control is not merely a theoretical concept, it becomes essential for achieving efficient, reliable operation and energy harvesting.

In this work, we derived exact optimal control protocols for systems driven by arbitrary time-dependent forces, using tools from stochastic thermodynamics and optimal control theory. We showed that the quasistatic work naturally decomposes into three contributions: (i) work from initial-state information, (ii) work due to slow dragging under time-averaged forces, and (iii) additional work enabled by responding to fast dynamical modes. When the time-integrated variance of the driving forces decays slowly, the extracted work can grow unboundedly as the protocol duration is increased. We have demonstrated that slow protocols allow the driving forces to place the particle into a high-energy state, before the trap subsequently extracts this energy.

We demonstrated the versatility of our framework through two key examples: periodic forcing and active self-propulsion. In periodic systems, work scales linearly with protocol duration. In active matter, we showed that finite angular correlations enhance work extraction, with the work increasing monotonically with the persistence time. We further compared quasistatic work bounds across several active particle models.

These principles have promising implications for the design of next-generation microscopic robots, such as those envisioned for

medical applications like targeted drug delivery[67,68]. Within the body, microscale machines are likely to encounter non-equilibrium environments, ranging from pulsatile blood flow to persistent molecular noise in the intercellular matrix, where harnessing ambient fluctuations could enhance performance and enable self-powered operation. Advances in experimental techniques, including optical tweezers and microfluidic systems, now make it possible to test optimal control strategies in such finite-time, fluctuating environments.

An intriguing avenue for future work is the application of similar optimization methods to settings with spatially varying forces, such as linear hydrodynamic flows, where analytical progress may still be feasible[69]. More generally, bounding the possible work extraction is critical for developing energy-efficient, adaptive control strategies for micro- and nanoscale technologies. Recent studies[70,71] have explored this problem from multiple perspectives, including the optimization of reactions in Markovian systems under topological, kinetic, and thermodynamic constraints, and the role of state preparation in setting bounds on extractable work. The optimal protocols derived in this work could also be applied repeatedly, aiding in non-equilibrium engines designs similar to those recently proposed for active matter[55].

Finally, our results highlight the role of non-equilibrium forces and fluctuations in enabling work extraction, drawing parallels with biological systems that naturally operate far from equilibrium. By identifying key constraints and providing insight into the performance and tradeoffs of work extraction protocols under realistic conditions, our work establishes a theoretical framework with relevance across both biological and technological systems.

## Methods

The optimal protocol discussed in the "Results" section was derived using Euler–Lagrange methods. Here, we derive the optimal protocol for arbitrary driving forces, as well as the associated mean particle trajectories and the thermodynamic work. We define the mean particle position as $\mathbf{q}(t) = \langle \mathbf{x}(t) \rangle$. The mean particle position evolves as

$$\gamma \dot{\mathbf{q}}(t) = -k[\mathbf{q}(t) - \boldsymbol{\lambda}(t)] + \mathcal{F}(t) \tag{37}$$

We let $\omega = k/\gamma$, so that the equation of motion takes the form

$$\dot{\mathbf{q}}(t) = -\omega[\mathbf{q}(t) - \boldsymbol{\lambda}(t)] + \mathcal{F}(t)/\gamma \tag{38}$$

For a harmonic control trap, we have the mean work

$$\mathcal{W} = k \int_0^{t_f} dt\, \dot{\boldsymbol{\lambda}}(t) \cdot [\boldsymbol{\lambda}(t) - \mathbf{q}(t)] \tag{39}$$

$$= \int_0^{t_f} dt\, \dot{\boldsymbol{\lambda}}(t) \cdot [\gamma \dot{\mathbf{q}}(t) - \mathcal{F}(t)] \tag{40}$$

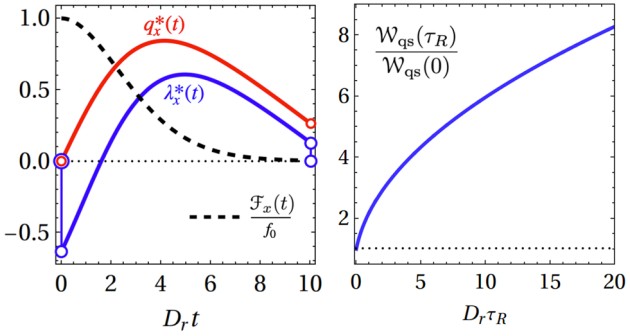

**Fig. 6 | Optimal protocol for an ABP starting and ending at the origin with** $\lambda_x(0) = \lambda_{x,f} = q_x(0) = 0$. Left: Optimal protocol trap position (solid blue line) and corresponding mean particle position (solid red line) for a persistent active Brownian particle with initial orientation along the positive x-axis. The dashed line indicates the amplitude of the force along the x-axis. The protocol features an initial backward jump, positioning the trap behind the particle. right: Quasistatic work $\frac{\mathscr{W}_{qs}(\mathfrak{D}_0)}{\mathscr{W}_{qs}(0)}$ as a function of the delay number $\mathfrak{D}_0$, demonstrating a monotonic increase. Parameters: $D_r\tau_R = 5$, $D_r t_f = 10$, $f_0 = \omega = \gamma = 1$.

**Table 1 | Various active particle models along with the quasistatic work extraction given by Eq. (10)**

| Active particle dynamics and quasistatic work extraction bound | | | | |
|---|---|---|---|---|
| Model | Fractional angular noise $\phi(t) \sim$ fBm($H$) | Random rotational diffusion $D_r \sim \frac{1}{\overline{D}} e^{-D_r/\overline{D}}$ | Chiral ABP $\dot{\phi}(t) = \omega_c + \sqrt{2D_r}\eta(t)$ | Accelerating ABP $\mathbf{f}(t) = f_0(t/\tau)^\alpha \hat{\mathbf{n}}(t)$ |
| Effective force $\mathscr{F}(t)$ | $f_0 \hat{\mathbf{n}}_0 e^{-D_H t^{2H}}$ | $\frac{f_0 \hat{\mathbf{n}}_0}{1 + \overline{D}t}$ | $f_0 \begin{bmatrix} \cos(\phi_0 + \omega t) \\ \sin(\phi_0 + \omega t) \end{bmatrix} e^{-D_r t}$ | $f_0 \hat{\mathbf{n}}_0 (t/\tau)^\alpha e^{-D_r t}$ |
| Quasistatic work | $\frac{\mathscr{W}_{qs}(H)}{\mathscr{W}_{qs}(1/2)} = \frac{D_{1/2}\Gamma(1 + \frac{1}{2H})}{2^{2H-1}D_H^{\frac{1}{2H}}}$ | $\mathscr{W}_{qs}(\overline{D} = D_r) = 2\mathscr{W}_{qs}^{ABP}$ | $\mathscr{W}_{qs} = \mathscr{W}_{qs}^{ABP}$ | $\frac{\mathscr{W}_{qs}}{\mathscr{W}_{qs}^{ABP}} = \frac{\Gamma(1 + 2\alpha)}{4^\alpha (D_r\tau)^{2\alpha}}$ |

Cases considered: 1) Fractional Brownian orientations characterized by a Hurst exponent $H$ such that $H \in (0, 1/2)$ gives anti-correlation reorientations, and $H \in (1/2, 1)$ correlation reorientations. At $H = 1/2$ normal ABP dynamics is recovered with $D_{1/2} = D_r$. 2) Random rotational diffusion, whereby the rotational diffusivity is random and exponentially distributed with mean $\overline{D}$. 3) Chiral ABPs where a mean angular velocity $\omega_c$ is included. 4) Accelerating ABPs where the self-propulsion force depends on time as a power-law with exponent $\alpha \in (-1/2, \infty)$. A direct comparison with the traditional ABP result $\mathscr{W}_{qs}^{ABP} = -f_0^2/(8\gamma D_r)$ is given in all cases.

where we used the equation of motion. Taking a further derivative of the mean equation of motion, we can also write

$$\dddot{\mathbf{q}}(t) = -\omega\ddot{\mathbf{q}}(t) + \omega\dot{\boldsymbol{\lambda}}(t) + \dot{\mathcal{F}}(t)/\gamma \tag{41}$$

which we can use to eliminate $\omega\dot{\boldsymbol{\lambda}}(t)$ in the expression for the work. This gives

$$\mathcal{W} = \frac{\gamma}{\omega}\int_0^{t_f} dt\left[\dddot{\mathbf{q}}(t) + \omega\ddot{\mathbf{q}}(t) - \dot{\mathcal{F}}(t)/\gamma\right]\cdot\left[\dot{\mathbf{q}}(t) - \mathcal{F}(t)/\gamma\right] \tag{42}$$

Many of these terms can be written as total time derivatives. For example, $\dot{\mathbf{q}}\cdot\ddot{\mathbf{q}} = \frac{1}{2}d(\dot{\mathbf{q}}^2)/dt$. Proceeding similarly, we have

$$\mathcal{W} = \frac{1}{2k}[(\gamma\dot{\mathbf{q}})^2(t)]_0^{t_f} + \frac{1}{2k}[\mathcal{F}^2(t)]_0^{t_f} - \frac{1}{\omega}[\dot{\mathbf{q}}(t)\cdot\mathcal{F}(t)]_0^{t_f} \tag{43}$$

$$+ \int_0^{t_f} dt\mathcal{L}(t,\mathbf{q},\dot{\mathbf{q}}) \tag{44}$$

where we introduced the Lagrangian $\mathcal{L}(t,\mathbf{q},\dot{\mathbf{q}}) = \gamma\dot{\mathbf{q}}(t)\cdot[\ddot{\mathbf{q}}(t) - \mathcal{F}(t)/\gamma]$. The corresponding Euler–Lagrange equation reads

$$\dddot{\mathbf{q}}(t) = \frac{\dot{\mathcal{F}}(t)}{2\gamma} \tag{45}$$

which can easily be solved by

$$\mathbf{q}(t) = \mathbf{q}_i + \boldsymbol{\varphi}t + \frac{1}{2\gamma}\int_0^t dt'\mathcal{F}(t') \tag{46}$$

where we used the initial condition $\mathbf{q}(0) = \mathbf{q}_i$. To proceed, we must identify the unknown constant $\boldsymbol{\varphi}$, which is typically done by minimizing the work with respect to $\boldsymbol{\varphi}$[19]. For this, we must first write the explicit expression for the work, which we do by using the equations of motion to determine the boundary conditions for the particle velocity:

$$\dot{\mathbf{q}}(0) = \omega(\boldsymbol{\lambda}_i - \mathbf{q}_i) + \mathcal{F}(0)/\gamma \tag{47}$$

$$\dot{\mathbf{q}}(t_f) = \omega(\boldsymbol{\lambda}_f - \mathbf{q}(t_f)) + \mathcal{F}(t_f)/\gamma \tag{48}$$

The work then explicitly reads

$$\begin{aligned}\mathcal{W} = &\frac{1}{2k}[(k(\boldsymbol{\lambda}_f - \mathbf{q}(t_f)) + \mathcal{F}(t_f))^2]\\ &-\frac{1}{2k}[(k(\boldsymbol{\lambda}_i - \mathbf{q}_i) + \mathcal{F}(0))^2] + \frac{1}{2k}[\mathcal{F}(t_f)^2 - \mathcal{F}(t_0)^2]\\ &-\frac{1}{k}[(k(\boldsymbol{\lambda}_f - \mathbf{q}(t_f)) + \mathcal{F}(t_f))\cdot\mathcal{F}(t_f)]\\ &+\frac{1}{k}[(k(\boldsymbol{\lambda}_i - \mathbf{q}_i) + \mathcal{F}(t_0))\cdot\mathcal{F}(t_0)]\end{aligned} \tag{49}$$

$$+\gamma\boldsymbol{\varphi}^2 t_f - \gamma\int_0^{t_f} dt\left(\frac{\mathcal{F}(t)}{2\gamma}\right)^2 \tag{50}$$

We note that there is here also a dependence on $\boldsymbol{\varphi}$ through terms containing $\mathbf{q}(t_f)$, from Eq. (46). From this expression, the unknown parameter $\boldsymbol{\varphi}$ can be determined by minimization $\partial_{\boldsymbol{\varphi}}\mathcal{W} = 0$, resulting in

$$\boldsymbol{\varphi}_* = \omega\frac{\boldsymbol{\lambda}_f - \mathbf{q}_i}{2 + \omega t_f} - \frac{\omega}{2 + \omega t_f}\int_0^{t_f} dt\frac{\mathcal{F}(t)}{2\gamma} \tag{51}$$

This gives the mean particle trajectory, which in turn through the equation of motion, can be used to find the optimal protocol. It takes

the form

$$\boldsymbol{\lambda}_*(t) = \boldsymbol{\lambda}_{eq}(t) + \boldsymbol{\lambda}_{neq}(t) \tag{52}$$

where

$$\boldsymbol{\lambda}_{eq}(t) = \mathbf{q}_i + \frac{1 + \omega t}{2 + \omega t_f}(\boldsymbol{\lambda}_f - \mathbf{q}_i) \tag{53}$$

and

$$\boldsymbol{\lambda}_{neq} = \int_0^t dt'\frac{\mathcal{F}(t')}{2\gamma} - \frac{1 + \omega t}{2 + \omega t_f}\int_0^{t_f} dt\frac{\mathcal{F}(t)}{2\gamma} - \frac{\mathcal{F}(t)}{2k} \tag{54}$$

Combining results, the work can be written

$$\begin{aligned}\mathcal{W} = &\frac{k}{2}[(\boldsymbol{\lambda}_f - \mathbf{q}(t_f))^2] - \frac{k}{2}[(\boldsymbol{\lambda}_i - \mathbf{q}_i)^2]\\ &+\left(k\frac{\boldsymbol{\lambda}_f - \mathbf{q}_i}{2 + \omega t_f} - \frac{k}{2 + \omega t_f}\int_0^{t_f} dt\frac{\mathcal{F}(t)}{2\gamma}\right)^2\frac{t_f}{\gamma}\end{aligned} \tag{55}$$

$$-\gamma\int_0^{t_f} dt\left(\frac{\mathcal{F}(t)}{2\gamma}\right)^2, \tag{56}$$

which is the general expression analyzed in the "Results" section.

## Data availability
No datasets were generated or analyzed during the current study.

## Code availability
No code was generated during the current study.

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

## Acknowledgements

K.S.O. acknowledges support from the Alexander von Humboldt Foundation. Y.R. acknowledges support from the Israel Science Foundation (grants No. 385/21). Y.R. and R.G. acknowledge support from the European Research Council (ERC) under the European Union's Horizon 2020 research and innovation program (Grant Agreement No. 101002392). R.G. acknowledges support from the Mark Ratner Institute for Single Molecule Chemistry at Tel Aviv University. H.L. acknowledges support by the Deutsche Forschungsgemeinschaft (DFG) within the project LO 418/29-1.

## Author contributions

K.S.O. conceived project. K.S.O., R.G., Y.R. and H.L. conducted research and contributed to the writing of the paper.

## Funding

## Competing interests

The authors declare no competing interests.
