## [Transparent Peer Review file · Nature Communications]

Harnessing non-equilibrium forces to optimize work extraction

Corresponding Author: Dr Kristian Olsen

Version 0:

Reviewer comments:

Reviewer #1

(Remarks to the Author)

Optimal control has emerged as a highly active research area within stochastic thermodynamics in recent years, focusing on the development of driving protocols that minimize thermodynamic costs such as work expenditure. This manuscript extends this line of research to systems experiencing time-dependent nonequilibrium forces.

The authors derive closed-form expressions for both the optimal protocol and the associated work, demonstrating that the total work naturally decomposes into three distinct contributions: (i) work arising from initial disequilibrium, (ii) work required to drag the system under time-averaged forces, and (iii) work induced by temporal fluctuations in the forces. The analysis is illustrated through two compelling examples -- a system driven by oscillating forces and an active Brownian particle. The authors also discuss how measurement errors impact work extraction efficiency.

The manuscript is exceptionally well-written and clear, the research is high quality, and the authors provide insightful physical interpretations. Generally, I recommend publication, with only a few minor comments.

First, I think the authors should be a bit clearer regarding the restrictive assumptions underlying the theoretical results. Consistent with most optimal control studies in this field, the authors assume that control is implemented exclusively through manipulation of a harmonic trap with fixed width. In addition, they assume that nonequilibrium forces depend only on time and not on position. While relaxing this latter assumption would likely preclude analytical solutions (due to coupling between the forces and protocol), it is a substantial limitation that warrants more prominent discussion. I would recommend that the authors emphasize these limitations and discuss the (im)possibility of generalizing the results to space-dependent forces.

Second, I am not really convinced by the interpretation of the optimal protocol as an "information engine". While the optimal protocol indeed exploits knowledge of the temporal structure of nonequilibrium forces, this information is provided a priori as input to the optimization algorithm. This differs from (my understanding of) an information engine, which should detect and respond to stochastic fluctuations in real-time during the process. That is, in genuine information engines, the driving protocol can vary depending on the stochastic trajectory realized by the system. By contrast, the authors' optimal protocol is predetermined through the optimization procedure before any trajectories are observed. I recommend that the authors de-emphasize their interpretation of the optimal protocol as an information engine, or explain it in more depth, to avoid confusion.

I did not find any typos but there is a typesetting issue near line 404.

Reviewer #2

(Remarks to the Author)

This work presents optimal protocols for harvesting energy from a force whose mean values at each time are known, using a particle in a harmonic potential or equivalent first-order dynamical system. Despite a few limitations, explored below, I think this makes a nice contribution and yields a number of insights that can guide future work.

Let's start with the strengths. The three-part decomposition into an information-based part and parts using time-integrated mean and variance of the force is pretty. The last term involving the integrated variance was surprising to me until I thought

about it more, and I think that's enough of an advance to favor publication. The other two terms are more familiar. The first has been explored in roughly equivalent studies of information engines and based on principles found, e.g., in Parrondo et al., [Ref. 30] and the second is similar to studies on optimal navigation (e.g. by the last author). Still, the deviation-based third term is a nice insight and sets one up to understand the application examples.

On the other hand, there are many limitations to the study:

- 1) Although Eq. 11 is a general result, almost all the discussion is couched in terms of quasistatic protocols, where the physical interpretation is perhaps clearest. But probably faster protocols are better (if not, why?).
- 2) The harvesting is the physicists' version, where the energy isn't really stored in any useful way but rather just amounts to "cooling the bath" This limits practical applications. I contrast the scheme here with previous (experimental) work from the Roichman group (Admon et al., PRL 2018) where the energy was captured (by having a particle move upstream in a flow field).
- 3) The force is rarely known, and one might wonder how useful results based heavily on this knowledge can be. The authors do address this in several ways: For example, in the section on periodic forcing with phase errors (not such a big deal) and frequency errors (a more serious problem). A systematic way to address such issues (measurements and then feedback) is hinted at in the second application, to active matter, which breaks with the rest of the paper by introducing a single measurement and feedback. One can think of each measurement as making possible a "burst" of harvested energy that decays exponentially (one has to piece this together by substituting the expression for σ_{ϕ}^2 into Eq. 25). The work acquired is finite (not proportional to τ_R), which is stated but not really emphasized. Of course, with repeated measurement and feedback, one could address this issue. But that is beyond the scope of this paper.

Given these limitations, I don't think it is fair for the authors to claim to have "uncover[ed] the fundamental limitations of energy harvesting" — that seems too strong. Nonetheless, they have identified some limitations and have provided a great deal of insight in work that is quite creative, and I would be in favor of its eventual publication in Nat. Commun.

Minor Points:

1. [I think this is a minor point, but it could be more important.] Figure 1 talks about "arbitrary time-dependent forces $\mathcal{F}(t)$. (The Discussion contains similar language.) Is this really the case? For example, could \mathcal{F} be a step function? Could the sine wave example have arbitrarily high frequencies? One imagines that for this case, there wouldn't be much to harvest if the external force is too fast relative to τ_R . Whatever the answer, the issue needs more discussion.
2. References 54 and 55 (preprints) cover very similar ground. There should be more discussion comparing that work and the present paper.
3. Ref. 26 was published in 2024 as a PRX paper.
4. Some other papers might warrant inclusion and at least a brief discussion:
 - i) A. Kolchinsky et al., "Maximizing Free Energy Gain", Entropy 2025.
 - ii) L. Gammaitoni et al., "Vibration Energy Harvesting..." from the book "Sustainable Energy Harvesting Technologies" Ed. Yen Kheng Tan, 2011.This second is a stand-in for a large group of papers from the "classical" world of energy harvesting, which is a well-developed engineering topic that has also had some contributions by physicists. It would be nice if the paper had more contact with this part of the literature.
5. Some of the paper exposition could be clearer. I did not see if it is stated that $W < 0$ corresponds to work extraction [if so, it is a bit buried].
6. Similarly, Eq. 12 would be clearer if the underbrace terms (W_i, W_{ta}, W_d) were immediately defined. For example, W_i is defined only in the caption of Fig. 2, which is one page after. In a first reading, I thought from Line 258 that the "i" in W_i was "initial", which led to some confusion.
7. Eq. 15: Define better the space \mathcal{B} of Boltzmann states. Is it the set of Boltzmann distributions for fixed temperature (the bath) but variable λ ? Related: The authors should show (perhaps in the supplement) how to derive Eq. 16. Given supplements, there seems no need to skip steps.

Reviewer #3

(Remarks to the Author)

The authors present a detailed theoretical investigation focusing on optimal work extraction through optimal control protocols applied to single-particle systems driven by time-dependent external forces. While the equilibrium component of these optimal protocols (Eq.7 in the manuscript) is well-established in the literature, the authors introduce an explicit analytical form for the nonequilibrium correction (Eq.8), which represents the main theoretical contribution of this work. Two concrete examples, periodic external driving forces (Case I) and active Brownian particles (Case II), are also provided. These examples clearly illustrate the derived optimal protocols and would be of interest to specialists working on stochastic thermodynamics.

From a technical perspective, the manuscript appears sound and clearly presented. However, I find that the overall novelty and broad interdisciplinary appeal is somewhat lacking. Specifically, the optimal control approach to work extraction in stochastic thermodynamics is already a well-explored topic at least theoretically. Seminal theoretical studies, such as those by Schmiedl and Seifert (PRL, 2007)[19], Sivak and Crooks (PRL, 2012), and more recent works including Blaber and Sivak (J. Phys. Comm., 2023), have extensively addressed this topic. Thus, while Eq.8 is theoretically novel and interesting, its overall conceptual contribution appears rather incremental. Furthermore, there is already an experimental demonstration of optimal control, Loos et al. (PRX, 2024)[26], and therefore the purely theoretical nature of the present manuscript would limit its impact potential for a journal targeting a broad interdisciplinary audience.

Additionally, the authors' characterization of their results as demonstrating an "information engine" is, in my view, not entirely convincing. Traditional information engines explicitly incorporate measurement and feedback control to extract work. While the authors acknowledge the absence of explicit measurement-feedback loops, their interpretation of "information" is primarily relevant to initial nonequilibrium states and precise knowledge of external forces. Thus, the use of the term "information engine" could be considered somewhat misleading.

In summary, although this manuscript presents theoretical results that will be of interest to specialists, its contributions appear somewhat incremental within a field already rich in theoretical insights. Therefore, while this study would be well suited for publication in a more specialized journal, I believe that it does not meet the threshold of broad impact and significant innovation required for publication in Nature Communications.

Version 1:

Reviewer comments:

Reviewer #1

(Remarks to the Author)

The authors have addressed my concerns. I recommend the manuscript for publication.

Reviewer #2

(Remarks to the Author)

I think the authors have done a good job in responding to my concerns and (as far as I can tell) those of the other reviewers. I particularly appreciated the new section on "Precision vs. risk", with its discussion of tradeoffs.

Regarding the general question of significance, it is important to be fair. That is, the original version of the paper was criticized for having made claims that are too strong. The claims have here been moderated to an appropriate level. But it would be a mistake to go too far the other way and dismiss the insights presented: as the authors state, the literature on optimal control of stochastic thermodynamic processes has been focused in recent years on minimizing dissipation. Here, the authors make a good case for using that formalism to discuss energy extraction, and then they establish first results that will lead to a program of future research on nonlinearities, faster protocols, etc. This contrasts with current "practical" efforts for energy harvesting, which are strongly linked to the idea of resonant structures interacting with known periodic forces. Although such a strategy overlaps at least a bit with "Case I", it does not work well in other situations such as "Case II". Because of this enlarged perspective, I think the paper deserves to be in Nature Communications.

Minor corrections:

Line 231: add ref. to SM Sec. 1

Line 235: quasistatic (typo)

Line 361: protocol's (typo)

Reviewer #3

(Remarks to the Author)

The authors have addressed the concerns raised in my previous report. In particular, the meaning of "information engine" is now clearer, and the manuscript has been revised satisfactorily. Nevertheless, I haven't been fully convinced about the level of conceptual novelty. For instance, in their response the authors state that "In particular, only in very few cases have optimal control protocols been used not merely to reduce dissipation, but to actively extract energy from nonequilibrium systems." However, when nonconservative forces are absent, reducing dissipation is exactly equivalent to enhancing work extraction. Thus, the novelty of this study appears to lie in the inclusion of external driving ("active") force, which is however restricted to be spatially homogeneous. The remaining question is whether this addition constitutes sufficient conceptual novelty for Nature Communications. Considering the other reports, my overall assessment is marginal: I do not offer a strong recommendation for publication, but I would not oppose acceptance if the other referees recommend.

Reply to reviewers: Harnessing non-equilibrium forces to optimize work extraction (NCOMMS-25-35338)

Dear Editors and referees,

We thank the reviewers for their careful reading of the manuscript, and for their constructive feedback. We also appreciate the positive comments, which includes stating that the manuscript is exceptionally well-written and clear, that the research is of high quality, and that it presents a number of insights that can guide future work.

Following the reviewers suggestions, the manuscript has improved in both clarity and relevance, and more clearly positions itself within the existing literature. We have also added substantial material, including a new section, that distinguishes our framework further from the existing ones and makes a clearer bridge to experiments. We hope that with these changes, the manuscript is suited for publication in Nature Communications.

Below we address individually the comments and questions raised by the referees. Our responses are highlighted in blue. Similarly, highlighted manuscript changes can be found in the attached markup PDF.

Best regards,

Kristian Stølevik Olsen, Rémi Goerlich, Yael Roichman and Hartmut Löwen.

Reviewer 1

Optimal control has emerged as a highly active research area within stochastic thermodynamics in recent years, focusing on the development of driving protocols that minimize thermodynamic costs such as work expenditure. This manuscript extends this line of research to systems experiencing time-dependent nonequilibrium forces.

The authors derive closed-form expressions for both the optimal protocol and the associated work, demonstrating that the total work naturally decomposes into three distinct contributions: (i) work arising from initial disequilibrium, (ii) work required to drag the system under time-averaged forces, and (iii) work induced by temporal fluctuations in the forces. The analysis is illustrated through two compelling examples – a system driven by oscillating forces and an active Brownian particle. The authors also discuss how measurement errors impact work extraction efficiency.

The manuscript is exceptionally well-written and clear, the research is high quality, and the authors provide insightful physical interpretations. Generally, I recommend publication, with only a few minor comments.

Response: We thank the reviewer for the summary of our work and for the positive evaluation of the manuscript.

First, I think the authors should be a bit clearer regarding the restrictive assumptions un-

derlying the theoretical results. Consistent with most optimal control studies in this field, the authors assume that control is implemented exclusively through manipulation of a harmonic trap with fixed width. In addition, they assume that nonequilibrium forces depend only on time and not on position. While relaxing this latter assumption would likely preclude analytical solutions (due to coupling between the forces and protocol), it is a substantial limitation that warrants more prominent discussion. I would recommend that the authors emphasize these limitations and discuss the (im)possibility of generalizing the results to space-dependent forces.

Response: We thank the reviewer for pointing out the restrictive assumptions underlying our analysis. As the reviewer notes, relaxing the second assumption would couple the external forces and the control protocol, and potentially make the problem non-linear, precluding closed-form analytical solutions. We have revised the manuscript to highlight this limitation more prominently by adding to the following in the Methods section:

"We emphasize that it is assumed that these driving forces are spatially-homogeneous, as other scenarios would render the problem non-linear and analytically untractable. Yet, a wide range of non-equilibrium systems can be captured by the above model, as we describe in future sections."

Relatedly, we draw attention to recent work by P. S. Pal et al. [arXiv:2504.09467], who studied engines in space-varying hydrodynamic flows. Although their setting does not yield optimal protocols, it illustrates possible directions for extending control strategies beyond the assumptions we adopt here. As a part of the revised discussion section, we mention;

"An intriguing avenue for future work is the application of similar optimization methods to settings with spatially varying forces, such as linear hydrodynamic flows, where analytical progress may still be feasible [69]."

Second, I am not really convinced by the interpretation of the optimal protocol as an "information engine". While the optimal protocol indeed exploits knowledge of the temporal structure of nonequilibrium forces, this information is provided a priori as input to the optimization algorithm. This differs from (my understanding of) an information engine, which should detect and respond to stochastic fluctuations in real-time during the process. That is, in genuine information engines, the driving protocol can vary depending on the stochastic trajectory realized by the system. By contrast, the authors' optimal protocol is predetermined through the optimization procedure before any trajectories are observed. I recommend that the authors de-emphasize their interpretation of the optimal protocol as an information engine, or explain it in more depth, to avoid confusion.

Response: We thank the reviewer for the comment. In the revised manuscript, we make it clear that we are not suggesting that the protocols are genuine information engines, but rather that there are some analogies worth exploring. We have removed the sentence claiming that the protocols act as automatic information engines, and have added the following discussion:

"Leveraging prior knowledge of the forces that drive the system away from equilibrium, Euler–Lagrange minimization gives rise to protocols that both anticipate and adapt to prescribed particle dynamics, enabling the spontaneous extraction of maximal energy from non-equilibrium processes."

We have also removed the term "information engine" from the section title dealing with pe-

riodic forces, as well as from the abstract and concluding discussion. Finally, we have re-written the paragraph comparing the optimal protocol with information engines in the case of periodic forces, which now reads:

"We see that for sufficiently slow protocols, compared to the forcing period, the optimal control is able to utilize the oscillations such that work can be extracted. The optimal protocol harnesses the dynamic information available and automatically extracts as much work as possible. This is a consequence of precise knowledge of the force at all times. The protocol consists of repeatedly letting the force move the particle into a high-energy state before shifting the potential accordingly to extract the stored energy as work. Let us recall that in Brownian information engines, feedback is used to rectify thermal noise and convert measurement information into work. In simple engine designs, the potential has the freedom to move left and right based on the measurement outcome, extracting work in a manner similar to the above protocols [47]. Figure (4) summarizes the main mechanism behind the work extraction in the case of periodic forces."

I did not find any typos but there is a typesetting issue near line 404.

Response: We thank the reviewer for the attention to detail. The typesetting error has been corrected in the revised manuscript, by splitting the equation into separate lines.

Reviewer 2

This work presents optimal protocols for harvesting energy from a force whose mean values at each time are known, using a particle in a harmonic potential or equivalent first-order dynamical system. Despite a few limitations, explored below, I think this makes a nice contribution and yields a number of insights that can guide future work.

Let's start with the strengths. The three-part decomposition into an information-based part and parts using time-integrated mean and variance of the force is pretty. The last term involving the integrated variance was surprising to me until I thought about it more, and I that's enough of an advance to favor publication. The other two terms are more familiar. The first has been explored in roughly equivalent studies of information engines and based on principles found, e.g., in Parrondo et al., [Ref. 30] and the second is similar to studies on optimal navigation (e.g. by the last author). Still, the deviation-based third term is a nice insight and sets one up to understand the application examples.

Response: We thank the reviewer for the positive evaluation and for highlighting the clarity and novelty of our decomposition. We are especially pleased that the term involving the integrated variance was found intriguing.

On the other hand, there are many limitations to the study:

1) Although Eq. 11 is a general result, almost all the discussion is couched in terms of quasistatic protocols, where the physical interpretation is perhaps clearest. But probably faster protocols are better (if not, why?).

Response: We thank the reviewer for this comment. Slower (quasistatic) protocols indeed allow for the extraction of more work per cycle, but at the cost of longer execution times. In contrast, finite-time protocols could be advantageous in situations where one aims to repeatedly implement the process, effectively achieving a higher rate of work extraction, even if the work

per cycle is somewhat reduced. This can equivalently be seen through thinking of the optimal protocol as an engine, where fixing the duration t_f fixes the power (work per time) as the ratio of work output and duration. Since the protocols are optimal for fixed t_f , we here consider the most efficient engines. In the context of periodic forces, as illustrated in Figure 3, we provide the intuition that protocols must be allowed sufficient time to exploit the temporal structure of the driving forces to be able to extract work.

2) The harvesting is the physicists' version, where the energy isn't really stored in any useful way but rather just amounts to "cooling the bath" This limits practical applications. I contrast the scheme here with previous (experimental) work from the Roichman group (Admon et al., PRL 2018) where the energy was captured (by having a particle move upstream in a flow field).

Response: We are aware of this important limitation. In our analysis, the extracted work could be viewed as an upper bound, representing the maximal amount of energy that could in principle be harvested if one also devised a suitable scheme or device for storing the energy. Indeed, the results we present could be used to transport a particle against a background force (like gravity), while simultaneously extracting work from the time-dependent forces. Our calculations therefore do not provide a ready-to-use energy harvesting protocol, but instead demonstrates the fundamental limit of how much work could be made available under these conditions.

3) The force is rarely known, and one might wonder how useful results based heavily on this knowledge can be. The authors do address this in several ways: For example, in the section on periodic forcing with phase errors (not such a big deal) and frequency errors (a more serious problem). A systematic way to address such issues (measurements and then feedback) is hinted at in the second application, to active matter, which breaks with the rest of the paper by introducing a single measurement and feedback. One can think of each measurement as making possible a "burst" of harvested energy that decays exponentially (one has to piece this together by substituting the expression for σ_ϕ^2 into Eq. 25). The work acquired is finite (not proportional to τ_R), which is stated but not really emphasized. Of course, with repeated measurement and feedback, one could address this issue. But that is beyond the scope of this paper.

Response: We agree that in realistic settings, the forces are rarely fully known. In the revised manuscript, we have added a new section (final section in "Methods" part) dealing with results that bounds the error in predicted work extraction arising from inaccuracies in force inference. This complements our existing analysis of amplitude and frequency errors and provides a more systematic framework for understanding the impact of imperfect force estimation.

Importantly, this analysis formulizes the intuitive picture of a tradeoff between the variability of the control protocol and the risk of error: erratic protocols with a large total variation can, in principle, exploit fine-scale details of the forces to extract more work, but they are also more susceptible to mistakes. Conversely, smoother protocols reduce the likelihood of errors but may leave some work unextracted. This insight provides guidance for tailoring control strategies depending on whether the priority is maximizing work extraction or minimizing the chance errors.

Finally, we emphasize that also for deterministic forces, an initial measurement may still be necessary. For example, in the case of periodic forces, a measurement may be required to determine the current phase and identify the optimal starting point for the protocol.

Given these limitations, I don't think it is fair for the authors to claim to have "uncover[ed] the fundamental limitations of energy harvesting" — that seems too strong. Nonetheless, they have identified some limitations and have provided a great deal of insight in work that is quite creative, and I would be in favor of its eventual publication in Nat. Commun.

Response: We thank the reviewer for the comment. We agree that the original wording was too strong, and we have revised the manuscript accordingly. In the revised manuscript, we have replaced "By uncovering the fundamental limits of energy harvesting from dynamic environments" with "By identifying key constraints and providing insight into the performance and tradeoffs of work-extraction protocols under realistic conditions". We hope this revised wording better reflects the scope of our work.

Minor Points:

1. [I think this is a minor point, but it could be more important.] Figure 1 talks about "arbitrary time-dependent forces $\mathcal{F}(t)$ ". (The Discussion contains similar language.) Is this really the case? For example, could \mathcal{F} be a step function? Could the sine wave example have arbitrarily high frequencies? One imagines that for this case, there wouldn't be much to harvest if the external force is too fast relative to τ_R . Whatever the answer, the issue needs more discussion.

Response: We thank the Reviewers for the comment. Regarding step-like forces, it is true that we assume smooth forces throughout, now emphasized in the methods section by using the sentence "An object, under the combined effect of a smooth external driving force" when introducing the forces. For the case of periodic forces (see for example Fig. 3), we observe that the period of the forces must be sufficiently short relative to the protocol duration to allow effective work extraction. This reflects the need for the protocol to adapt to the temporal variations in the forces. In the limit of very long protocols, Eq. (29) shows that the extracted work becomes independent of the periodicity of the forces. This can be understood directly from Eq. (15), where we see that the work in this limit is proportional to the time-integrated variance of the forces. For a simple periodic force, the variance is simply $1/2$, without any period dependence. Intuitively, rapid force oscillations cause small particle displacement and less work extraction, but also leads to more cycles of harvesting per unit time. We emphasize this in the revised manuscript by adding the following discussion in the section on periodic forces after noting that the work is independent of τ_p :

"This results from a direct competition between two effects caused by changing τ_p : firstly, it affects the amplitude of the particle displacements and hence how much energy could be harvested per cycle. Secondly, it also changes how many cycles of harvesting can be completed in a given total duration t_f . For slow protocols, these effects balance, resulting in Eq. (29)."

Eq. (32), however, shows that when there are measurement errors in the periodicity, the work extracted grows with the period of the force. Thus, the reviewer's intuition is correct and excessively rapid forces lead to reduced work extraction. The revised manuscript addresses this by including in the section on periodic forces:

"It is also worth emphasizing that when there are errors in the periodicity, the quasistatic work extraction becomes smaller when the periodicity τ_p is reduced. This reflects the fact that for very rapid forces, even small errors would make work extraction hard."

2. References 54 and 55 (preprints) cover very similar ground. There should be more discussion comparing that work and the present paper.

Response: In the revised manuscript, we mention the similarities and differences between our work and these references. We have also updated the references to their published versions. In the results section when analyzing work extraction from different active matter classes, we now include the following discussion:

"Recently, repeating this scheme was proposed as a way of constructing active information engines [55,56]. In the case of one-dimensional run-and-tumble or active Ornstein-Uhlenbeck particles, we recover the results of [55], while our general results can also be used to study a wide range of other active matter models, also in higher dimensions."

3. Ref. 26 was published in 2024 as a PRX paper.

Response: The reference has been updated.

4. Some other papers might warrant inclusion and at least a brief discussion: i) A. Kolchinsky et al., "Maximizing Free Energy Gain", Entropy 2025. ii) L. Gammaitoni et al., "Vibration Energy Harvesting..." from the book "Sustainable Energy Harvesting Technologies" Ed. Yen Kheng Tan, 2011. This second is a stand-in for a large group of papers from the "classical" world of energy harvesting, which is a well-developed engineering topic that has also had some contributions by physicists. It would be nice if the paper had more contact with this part of the literature.

Response: We thank the referee for pointing this out. To clarify, the work by Gammaitoni et al. (Ref 45) is cited in the section on oscillatory forces. We have re-written the introductory discussion that situates our results relative to the body of work on classical harvesting methods from oscillatory forces:

"Many energy harvesting solutions are based on periodic forces or motion, such as wave-energy converters, wearable fabrics that extract energy from movement, and piezoelectric generators that can charge pacemakers through heartbeats [5,6,9,45,46]. These classical approaches typically utilize mechanical oscillators coupled to transducers, aiming to convert displacement, velocity, or strain into electrical power. Here we consider a simple microscopic analogy, which in contrast relies on optimal control to identify limits and optimal strategies for energy harvesting, independent of any specific transducer design."

Similarly, while the work by Kolchinsky et al. was already mentioned in our concluding discussion, where we now mention that

"More generally, bounding the possible work extraction is critical for developing energy-efficient, adaptive control strategies for micro- and nanoscale technologies. Recent studies [70,71] have explored this problem from multiple perspectives, including the optimization of reactions in Markovian systems under topological, kinetic, and thermodynamic constraints, and the role of state preparation in setting bounds on extractable work."

5. Some of the paper exposition could be clearer. I did not see if it is stated that $W < 0$ corresponds to work extraction [if so, it is a bit buried].

Response: We are grateful for the comment, and have corrected this oversight in the revised version. When defining the work, we now state:

"We use the convention that a positive work implies energy has to be paid, while negative work corresponds to energy extracted."

6. Similarly, Eq. 12 would be clearer if the underbrace terms (W_i, W_{ta}, W_d) were immediately defined. For example, W_i is defined only in the caption of Fig. 2, which is one page after. In a first reading, I thought from Line 258 that the "i" in W_i was "initial", which led to some confusion.

Response: We thank the reviewer for raising this point. In the revised manuscript, we now define these terms more explicitly, stating:

"Taking the slow limit ($t_f \rightarrow \infty$) of Eq. (11) we find

$$\mathcal{W}_{qs} = \mathcal{W}_i + \mathcal{W}_{ta} + \mathcal{W}_d, \quad (\text{R1})$$

where the work has been decomposed into three parts; an information theoretic contribution, a contribution from time-averaged forces, and a contribution from forces deviating from its time-average, respectively taking the form

$$\mathcal{W}_i = -\frac{1}{2}k(\boldsymbol{\lambda}_i - \mathbf{q}_i)^2, \quad (\text{R2})$$

$$\mathcal{W}_{ta} = -(\mathbf{q}_f - \mathbf{q}_i) \lim_{t_f \rightarrow \infty} \overline{\mathcal{F}}(t_f), \quad (\text{R3})$$

$$\mathcal{W}_d = -\frac{1}{4\gamma} \lim_{t_f \rightarrow \infty} t_f \text{Var}(\mathcal{F}; t_f). \quad (\text{R4})$$

7. Eq. 15: Define better the space \mathcal{B} of Boltzmann states. Is it the set of Boltzmann distributions for fixed temperature (the bath) but variable lambda? Related: The authors should show (perhaps in the supplement) how to derive Eq. 16. Given supplements, there seems no need to skip steps.

Response: We thank the reviewer for raising this point. In the revised manuscript, we now define these terms more explicitly, stating:

"This operation projects the initial distribution $p_i(\mathbf{x})$ onto the space $\mathcal{B} = \{Z^{-1}e^{-\beta V[\mathbf{x}, \boldsymbol{\mu}]}, \boldsymbol{\mu} \in \mathbf{R}\}$ of Boltzmann states compatible with the trap manipulation—here, fixed-variance (i.e., constant β) Gaussian densities with variable location $\boldsymbol{\mu}$ (often called shift measures)."

We have also provided a detailed derivation of the result $k_B T D_{\text{KL}}(\pi[p_i] \parallel p_{\text{eq}}) = \frac{1}{2}k(\boldsymbol{\lambda}_i - \mathbf{q}_i)^2$ in the Supplemental Information, under the name "Derivation of Eq. (19)" in the revised version.

Reviewer 3

The authors present a detailed theoretical investigation focusing on optimal work extraction through optimal control protocols applied to single-particle systems driven by time-dependent external forces. While the equilibrium component of these optimal protocols (Eq.7 in the manuscript) is well-established in the literature, the authors introduce an explicit analytical form for the

nonequilibrium correction (Eq.8), which represents the main theoretical contribution of this work. Two concrete examples, periodic external driving forces (Case I) and active Brownian particles (Case II), are also provided. These examples clearly illustrate the derived optimal protocols and would be of interest to specialists working on stochastic thermodynamics.

From a technical perspective, the manuscript appears sound and clearly presented. However, I find that the overall novelty and broad interdisciplinary appeal is somewhat lacking. Specifically, the optimal control approach to work extraction in stochastic thermodynamics is already a well-explored topic at least theoretically. Seminal theoretical studies, such as those by Schmiedl and Seifert (PRL, 2007)[19], Sivak and Crooks (PRL, 2012), and more recent works including Blaber and Sivak (J. Phys. Comm., 2023), have extensively addressed this topic. Thus, while Eq.8 is theoretically novel and interesting, its overall conceptual contribution appears rather incremental. Furthermore, there is already an experimental demonstration of optimal control, Loos et al. (PRX, 2024)[26], and therefore the purely theoretical nature of the present manuscript would limit its impact potential for a journal targeting a broad interdisciplinary audience.

Response: We thank the reviewer for the positive evaluation of our work as technically sound and clearly presented. We also appreciate the constructive comments regarding novelty and scope. We would like to clarify that the primary aim of this manuscript is not simply to present another optimal protocol. Rather, we seek to highlight a shift in perspective: optimal control can be leveraged not only to minimize dissipation or implement control at lower energetic cost, but also to actively extract useful work from non-equilibrium driving. In this way, our study reframes the role of optimal control within stochastic thermodynamics.

In the revised version of the manuscript, we have made substantial additions to emphasize this perspective. In particular, we now include a new section that quantifies the errors associated with force inference and explores the resulting tradeoffs between precision and reliability in work extraction. These additions further underscore the broader conceptual implications of our framework beyond the derivation of a single protocol, as well as making it more experimentally relevant.

Additionally, the authors' characterization of their results as demonstrating an "information engine" is, in my view, not entirely convincing. Traditional information engines explicitly incorporate measurement and feedback control to extract work. While the authors acknowledge the absence of explicit measurement-feedback loops, their interpretation of "information" is primarily relevant to initial nonequilibrium states and precise knowledge of external forces. Thus, the use of the term "information engine" could be considered somewhat misleading.

Response: We appreciate the reviewer's thoughtful comment regarding our characterization of the results as an "information engine." As noted by the other reviewers as well, we recognize that our use of the term may be misleading, since traditional information engines explicitly involve measurement and feedback. To avoid confusion, we have revised the manuscript accordingly (see our detailed response to Reviewer 1, clarifying the distinction and softening our use of the term "information engine" when discussing our protocols).

In summary, although this manuscript presents theoretical results that will be of interest to specialists, its contributions appear somewhat incremental within a field already rich in theoretical insights. Therefore, while this study would be well suited for publication in a more

specialized journal, I believe that it does not meet the threshold of broad impact and significant innovation required for publication in Nature Communications.

Response: We thank the reviewer for their careful evaluation. We would like to clarify a few points in response to the concern about novelty and impact:

- While it is true that the underlying mathematical tools (e.g., Euler–Lagrange minimization) are classical, the perspective and context we develop here are new. In particular, only in very few cases have optimal control protocols been used not merely to reduce dissipation, but to actively extract energy from nonequilibrium systems. Our work highlights and formalizes this perspective, which is a field of growing interest.
- Following the reviewers' feedback, we have significantly extended the manuscript. These additions broaden the scope and strengthen the conceptual contribution of the study.
- We believe the target audience *is* broad. The work lies at the intersection of several fields, with relevance to thermodynamics and statistical physics, control theory and applied mathematics, biological systems such as molecular motors that optimally move according to non-equilibrium fluctuations, and engineering applications including robotics and micro-devices. By bridging these areas, the manuscript speaks to an interdisciplinary readership beyond a specialist audience, making Nature Communications an ideal arena.

We hope that the clarifications and revisions we have made will lead the reviewer to reconsider the manuscript's novelty, interdisciplinary appeal, and suitability for publication.

Reply to reviewers: Harnessing non-equilibrium forces to optimize work extraction (NCOMMS-25-35338)

Dear Editors and referees,

We thank the reviewers for carefully reading the revised manuscript and for the final comments. All minor suggestions has been incorporated in the revised manuscript.

Best regards,

Kristian Stølevik Olsen, Rémi Goerlich, Yael Roichman and Hartmut Löwen.

Reviewer 1

The authors have addressed my concerns. I recommend the manuscript for publication.

Response: We thank the reviewer for the positive evaluation.

Reviewer 2

I think the authors have done a good job in responding to my concerns and (as far as I can tell) those of the other reviewers. I particularly appreciated the new section on “Precision vs. risk”, with its discussion of tradeoffs.

Regarding the general question of significance, it is important to be fair. That is, the original version of the paper was criticized for having made claims that are too strong. The claims have here been moderated to an appropriate level. But it would be a mistake to go too far the other way and dismiss the insights presented: as the authors state, the literature on optimal control of stochastic thermodynamic processes has been focused in recent years on minimizing dissipation. Here, the authors make a good case for using that formalism to discuss energy extraction, and then they establish first results that will lead to a program of future research on nonlinearities, faster protocols, etc. This contrasts with current “practical” efforts for energy harvesting, which are strongly linked to the idea of resonant structures interacting with known periodic forces. Although such a strategy overlaps at least a bit with “Case I”, it does not work well in other situations such as “Case II”. Because of this enlarged perspective, I think the paper deserves to be in Nature Communications.

Response: We are grateful for the positive evaluation, and for highlighting the novelty of our work. We are glad that the reviewers shares our conviction that energy extraction is an interesting direction in optimal thermodynamic control.

Minor corrections:

Line 231: add ref. to SM Sec. 1

Line 235: quasistatic (typo)

Line 361: protocol's (typo)

Response: The reference to the Supplementary Information has been added, and typos have been corrected.

Reviewer 3

The authors have addressed the concerns raised in my previous report. In particular, the meaning of "information engine" is now clearer, and the manuscript has been revised satisfactorily. Nevertheless, I haven't been fully convinced about the level of conceptual novelty. For instance, in their response the authors state that "In particular, only in very few cases have optimal control protocols been used not merely to reduce dissipation, but to actively extract energy from nonequilibrium systems." However, when nonconservative forces are absent, reducing dissipation is exactly equivalent to enhancing work extraction. Thus, the novelty of this study appears to lie in the inclusion of external driving ("active") force, which is however restricted to be spatially homogeneous. The remaining question is whether this addition constitutes sufficient conceptual novelty for Nature Communications. Considering the other reports, my overall assessment is marginal: I do not offer a strong recommendation for publication, but I would not oppose acceptance if the other referees recommend.

Response: We are glad that the review thinks we have addressed the concerns in the previous report, and thank the reviewer again for pushing us to clarify the conceptual issue relating to information engines.